

# A Simplified Chemistry-Dynamical Model

Hao-Jhe Hong[1,2], Thomas Reichler[1]

[1]Department of Atmospheric Sciences, University of Utah, Salt Lake City, 84112, USA
[2]Research Center for Environmental Changes, Academia Sinica, Taipei City, 11529, Taiwan

*Correspondence to*: Hao-Jhe Hong (haojhe.hong@utah.edu)

**Abstract.** Recent observational and modeling studies show that variations of stratospheric ozone and the resulting interaction between ozone and the stratospheric circulation play an important role for surface weather and climate. However, in many cases computationally expensive coupled chemistry models have been used to study these effects. Here, we demonstrate how a much simpler idealized general circulation model (GCM) can be used for studying the impact of

interactive stratospheric ozone on the circulation. The model, named simplified chemistry-dynamical model (SCDM), is constructed from a preexisting idealized GCM, into which a simplified linear ozone scheme and a parameterization for the shortwave radiative effects of ozone are implemented. The distribution and variability of stratospheric ozone simulated by the new model are in good agreement with the MERRA2 reanalysis, even for extreme circulation events such as Arctic stratospheric sudden warmings. The model thus represents a promising new tool for the study of ozone-circulation

interaction in the stratosphere and its associated effects on tropospheric weather and climate.

## 1 Introduction

Idealized models are becoming increasingly popular for the study of phenomena that are too difficult to understand with more comprehensive models. The idea behind idealized models is to not represent every detail of the system, and instead only include processes that are relevant for the phenomenon to be explained (Held, 2005; Polvani et al., 2017; Strevens,

2017). The simplified environment then helps to isolate, study, and understand the behavior of the relevant processes. The relative simplicity also makes these models cheap to run, so that they are frequently used to explore the parameter sensitivities of specific processes, in which a large number of simulations with slightly different parameter settings are carried out.

Idealized models in the atmospheric sciences have a relatively long history. One widely used class of models are simplified

dry dynamical cores, which solve the primitive equations on a sphere to study the flow in a rotating atmosphere. Held and Suarez (1994) proposed a commonly used set-up for a dry dynamical core, in which the primitive equations are forced by a Newtonian relaxation term that nudges temperatures toward a given profile, with a Rayleigh damping term to mimic the momentum removal at the surface. Since the original proposal, dry dynamical cores have been expanded in many ways to address a wide range of problems. Examples include the addition of a stratosphere to investigate the dynamical coupling

between the stratosphere and troposphere (Polvani and Kushner, 2002), the implementation of moisture and gray radiation to



focus on the interplay between latent heat release and the dynamics (Frierson et al., 2006), or the incorporation of an idealized (Gerber and Polvani, 2009) or actual topography (Wu and Reichler, 2018) to generate planetary waves. Some of the most recent developments like "Isca" by Vallis et al. (2018) or "EMIL" by Garny et al. (2020) are modular systems, which give users a choice of parameterizations to simulate the global atmosphere at varying levels of complexity.

The present study is motivated by the need to better understand the role of ozone for the dynamics of the atmosphere. Ozone has long been realized as an important atmospheric trace gas that influences the trend of the Southern Annular Mode (SAM) and the strength of the Antarctic polar vortex through its radiative heating (Gillet and Thompson, 2003; Randel and Wu, 1999; Seviour et al., 2017; Son et al., 2010; Thompson and Solomon, 2002). The radiative effects of ozone also modulate the propagation and the breaking of planetary waves and the strength of Arctic polar vortex in the northern hemisphere (NH)

winter (Albers and Nathan, 2012; Nathan and Cordero, 2007). However, there still remain many unanswered questions, for example about the existence and the nature of feedbacks between ozone and the circulation, and how important these feedbacks are. Some modelling studies have already suggested a stronger stratosphere-troposphere interaction when interactive ozone is introduced in their simulations (Haase and Matthes, 2019; Li et al., 2016; Lin and Ming, 2021; Lin et al., 2017; Romanowsky et al., 2019), but these studies were based on complex chemistry-climate models that have a large

computational burden and are often difficult to understand. We believe that a simpler, more idealized approach can also be used to study the problem if the underlying idealized model is capable of simulating the basic processes behind the ozone-dynamics interaction, including a realistic circulation, the transport of ozone by the circulation, the photochemical processes of ozone, and the radiative impact of ozone on the circulation.

      This paper represents the first step towards studying the influence of interactive ozone under a simplified modelling

framework. We describe the construction and validation of an idealized model that simulates - in simplified ways - the interaction between ozone and the stratospheric circulation. The new model, denoted Simplified Chemistry-Dynamical Model (SCDM), is an extension of the dry dynamical core setup by Wu and Reichler (2018) (hereafter WR18). The model includes a simplified photochemical ozone scheme and a shortwave radiation parameterization for ozone. As we will show, the model creates realistic simulations of the global circulation, ozone distribution, and the diabatic heating by ozone, and

the model produces faithful life-cycle composites of stratospheric sudden warming events (SSWs). This underlines the potential of the model to study the role of ozone for the dynamical variability of the atmosphere.

      The paper is structured as follows. Section 2 provides details of the model setup, ozone scheme, shortwave parameterization, and design of the simulations. Section 3 describes the diabatic forcing of the model. Section 4 presents the model climatologies in terms of the global circulation and ozone and analyzes the circulation and ozone responses to SSWs. A

summary and an outlook are provided in Section 5.



## 2 Simplified Chemistry-Dynamical Model (SCDM)

SCDM consists of an idealized general circulation model (GCM), a simplified linear photochemical ozone scheme, and a shortwave parameterization for the radiative effects of ozone. The GCM treats ozone as a passive tracer to simulate the 3-dimensional dynamical ozone transport, and receives additional photochemical ozone tendency updates from the ozone

scheme. The temperatures are influenced by the dynamics and a Newtonian relaxation term, and also by the shortwave radiation parameterization scheme that considers the distribution of ozone. The radiation creates a feedback of ozone on temperatures and thus on the dynamics, accomplishing our goal to simulate the interaction between ozone and the dynamics. The source code of the model can be obtained at https://doi.org/10.5281/zenodo.4780888.

### 2.1 Idealized GCM

The idealized model is based on the Geophysical Fluid Dynamics Laboratory's (GFDL) spectral dynamical core at a horizontal resolution of T42 and 40 hybrid levels (Polvani and Kushner, 2002), with a model top at 0.01 hPa. Following Held and Suarez (1994), the model is driven by a seasonally varying Newtonian relaxation term

$$Q = -\frac{T - T_{eq}}{\tau}, \tag{1}$$

where $Q$ represents the diabatic forcing calculated as the difference between temperature $T$ and a prescribed equilibrium

temperature $T_{eq}$, divided by a relaxation time $\tau$ as in Jucker et al. (2014). The horizontal wind $v$ is damped near the boundary layer using Rayleigh friction of the form

$$\frac{\partial v}{\partial t} = -k_v(\sigma)v, \tag{2}$$

where $\sigma$ denotes the model's vertical sigma level. The damping rate $k_v$ decreases linearly with height from a surface value $k_f$ and has the form:

$$k_v = k_f \, max\left(0, \frac{\sigma - \sigma_b}{1 - \sigma_b}\right), \tag{3}$$

where $\sigma_b = 0.7$. WR18 showed that a surface drag of $k_f = 1.35$ results in good simulations of the zonal wind and the frequency of SSWs. We also use a realistic bottom topography derived from the GFDL AM2.1 climate model (Anderson et al., 2004) and a zonally asymmetric and seasonally varying $T_{eq}$ as in WR18.

### 2.2 Ozone scheme

We use version 2.9 of the simplified linear ozone scheme of Cariolle and Teyssèdre (2007), which has been widely used in similar previous studies with numerical models (e.g., Monge-Sanz et al., 2021). The scheme was initially developed by Cariolle and Déqué (1986) from a two-dimensional photochemical model, and has since then been widely used in studies with simpler GCMs and chemical transport models. Cariolle and Teyssèdre (2007) improved the original scheme by recalculating the necessary coefficients using more accurate representations of chemical reaction rates and dynamical

transport processes, and by including the effects of heterogeneous ozone chemistry.



The scheme calculates the photochemical ozone tendencies from a first-order (linear) Taylor expansion associated with the local deviations from climatology of (1) the volume ozone mixing ratio $r_{O_3}$, (2) the temperature $T$, and (3) the column amount of ozone $\Sigma O_3$ above the current pressure level, with an extra independent term that describes ozone destruction by heterogeneous chemistry

$$\frac{dr_{O_3}}{dt} = c_0 + c_1(r_{O_3} - \overline{r_{O_3}}) + c_2(T - \bar{T}) + c_3(\Sigma O_3 - \overline{\Sigma O_3}) + c_4 r_{O_3}. \tag{4}$$

Overbars denote relaxation climatologies, which in the SCDM are monthly mean three-dimensional climatologies from MERRA2 reanalysis (Bosilovich et al., 2015) over 1980-2018. $c_0$ - $c_4$ are monthly and spatially varying coefficients that represent the chemical reaction rates associated with each process, derived from off-line calculations with a chemistry model. $c_4$ is associated with the heterogeneous ozone destruction, for which Cariolle and Teyssèdre (2007) provide two prescriptions for assigning a non-zero value to it: one requires the instantaneous local temperatures to reach the critical threshold for the formation of polar stratospheric cloud ($T < 195K$), while the other one additionally includes a cold tracer that considers the movement of the cold air mass (see their study for details). We use the latter set-up for our SCDM.

Fig. 1 compares the relaxation time $\tau$ (fixed in time) of the Newtonian forcing (Eq. 1) (Fig. 1a) with the chemical relaxation time for ozone (for February) (Fig. 1b), given by ($-1/c_1$) (Eq. 4). Below 10 hPa, the chemical relaxation time for ozone is much longer than the Newtonian time scale, demonstrating that in this region of the atmosphere the distribution of ozone is largely controlled by transports from the dynamics and that photochemical processes play only a minor role here. The situation reverses in the upper stratosphere, where the chemical time scale is much shorter than the Newtonian one.

## 2.3 Radiative parameterization

We implement the shortwave parameterization by Lacis and Hansen (1974) into the SCDM to describe the absorption of solar radiation by ozone in the ultraviolet ($\lambda \lesssim 0.35 \, \mu$m) and visual ($0.5 \, \mu$m $\lesssim \lambda \lesssim 0.7 \mu$m) parts of the spectrum. The parameterization is based on accurate multiple-scattering computations that consider the amount of clouds, humidity, solar zenith angle, surface albedo, and vertical distribution of ozone. Since the SCDM is based on a dry dynamical core, we use the prescription for clear sky conditions. The surface albedo is obtained from the daily MERRA2 climatology (1980-2018). The shortwave parameterization calculates the fraction of absorbed total solar flux $A_l$ at model level $l$, and the resulting temperature tendency is calculated using

$$\frac{\Delta T}{\Delta t} = \frac{S g A_l}{c_p \Delta p \Delta t}, \tag{5}$$

where $S$ is the incoming solar flux, and the remaining quantities follow standard notation. $S$ varies with the Earth-Sun distance ($d$) and thus with the day of year $d_n$. It is calculated from $S = (\bar{d}/d)^2 S_o$, where $S_o$ = 1361 W m$^{-2}$ is total solar irradiance, and $\bar{d}/d$ represents variations in the Earth-Sun distance estimated by

$$\left(\frac{\bar{d}}{d}\right)^2 = \sum_{n=0}^{2} a_n \cos\left(n\frac{2\pi d_n}{365}\right) + b_n \sin\left(n\frac{2\pi d_n}{365}\right), \quad d_n = 0, 1, \dots, 364. \tag{6}$$

The coefficients are $a_n$ = [1.000110, 0.034221, 0.000719] and $b_n$ = [0.0, 0.001280, 0.000077], taken from Hartmann (2016).





## 2.4 Simulation set-up

The starting point for our simulations is the zonally and monthly varying $T_{eq}$ from WR18 to represent the Newtonian forcing (Eq. 1). However, the $T_{eq}$ from WR18 also represents the effects from ozone shortwave heating, which must be removed

because ozone heating is explicitly represented in the SCDM. Following WR18, this is achieved through an iterative technique that minimizes the difference between the simulated and observed temperature climatology according to

$$T_{eq,(N+1)} = T_{eq,(N)} - \frac{2}{3}\left(\bar{T}_{(N)}^{Y} - \bar{T}\right), \quad N = 1, 2, \dots 29, \quad Y = 4 \dots 500. \tag{7}$$

Here, $\bar{T}_{(N)}^{Y}$ indicates the monthly varying three-dimensional model temperature climatology from the $N^{th}$ iteration which is $Y$-years-long (with the first year discarded because of spin-up), and $\bar{T}$ is the MERRA2 temperature climatology as in Eq. (4).

We start with $T_{eq}$ from WR18 and integrate only for $Y = 4$ years to prevent the build-up of unrealistically high temperatures as the model is forced initially by both the uncorrected $T_{eq}$ and the ozone scheme. For the following iterations, we gradually increase $Y$ from 50 to 500, and end after 29 iterations as the model converges towards a realistic temperature climatology. The new $T_{eq}$ implicitly represents the effects from all diabatic heating sources, except the one associated with ozone shortwave heating, as this is explicitly represented in the model. The overall diabatic heating rate of the SCDM is then $Q =$

$Q_{newtonian} + Q_{ozone}$. The updated $T_{eq}$ is used to perform a 2000-years-long control run, needed to derive the necessary climatologies.

## 2.5 Dynamics diagnostics

To diagnose the upward propagating planetary wave activity, we use the vertical component of quasi-geostrophic Eliassen-Palm flux ($F_p$; Andrews et al., 1987), given by

$$F_p = -a\cos\phi f \frac{\overline{v'\theta'}}{\theta_p}, \tag{8}$$

where all symbols are standard notations. To validate the simulated climatologies against the MERRA2 reanalysis we use a two-tailed Student's $t$ test at the 95% confidence level.

## 3 Model forcing

In this section we validate the model's shortwave parameterization scheme and the structure of its diabatic forcing.

## 3.1 Shortwave ozone heating

We test the SCDM shortwave parameterization by overriding the model's internally generated ozone with daily MERRA2 ozone. We perform an ensemble of 39 one-year-long simulations, using MERRA2 ozone from years 1980-2018 as external input to the model's shortwave heating scheme. We then compare the daily mean shortwave heating rate from the 39 independent simulations with that from the MERRA2 climatology. Since the shortwave heating only related to ozone is not





available from MERRA2, the temperature tendency due to shortwave radiation under clear sky conditions is used as a proxy for it. This is a reasonable approximation in the stratosphere, where ozone is the dominant shortwave absorber. Fig. 2 compares the resulting January-March zonal mean shortwave heating rates from MERRA2 and the SCDM. The two fields are fairly similar, except in the lower troposphere and upper stratosphere. The underestimation by SCDM of 0.5-0.7 K/day in the upper stratosphere can be attributed to trace gases other than ozone, such as oxygen. Likewise, the underestimation in the
tropical lower troposphere is probably due to the dominant role of water vapor for the shortwave absorption in this region, which is also not included in the model's shortwave scheme.

### 3.2 Model diabatic heating

To further test the SCDM and its simple physical parameterizations, we present in Fig. 3 the January-March mean vertically integrated diabatic heating ($Q_{clm}$) in the stratosphere (1-150 hPa) and the troposphere (150-1000 hPa), for MERRA2 and the
2000-years-long run with the SCDM. Mathematically, the calculations can be written as follows:

$$Q_{clm} = \int_{p1}^{p2} Q \, dp / (p_2 - p_1),\qquad\qquad(9)$$

where $Q$ indicates the local diabatic heating at pressure level $p$. The MERRA2 temperature tendency due to physics is used to validate the diabatic heating of the SCDM.

As demonstrated by WR18, using zonally asymmetric $T_{eq}$ improves the structure of the tropospheric diabatic heating in an
idealized model, and indeed, our model's diabatic heating in the troposphere agrees fairly well with MERRA2 (Fig. 3, bottom), in particular in a zonal mean sense (Fig. 3c) of the middle and high latitudes. However, there are also some discrepancies, mostly in the tropical troposphere, and also in the stratosphere. WR18 explained the too small diabatic heating in the tropical troposphere from the unrealistic representation of convection and latent heat release in an idealized model. The details of the diabatic heating discrepancies in the stratosphere between MERRA2 and the model (Fig. 3, top), however,
are not well understood. We believe that the artifacts are mostly a consequence of the iterative procedure, that corrects small errors in the phasing and amplitude of the planetary waves to bring the simulated temperature structure of the model in agreement with MERRA2. We also note that the zonal mean heating between MERRA2 and the model are in rather good agreement.

### 4 Model validation

In this section, model climatologies from the 2000-years-long control simulation with the SCDM are presented and validated against the MERRA2 climatologies over 1980-2018. We note that the SCDM has by construction a quite realistic temperature climatology due to the relaxation towards the MERRA2 climatology (see Sect. 2.4).



### 4.1 Dynamical quantities

The vertical component of the Eliassen-Palm (EP) flux vector (Eliassen and Palm, 1961) is a common measure for the
upward propagating planetary wave activity from the troposphere. Due to its dominant role in driving the wintertime residual
mean circulation and the transport of ozone in the stratosphere, we begin our discussion on the vertical EP-flux component (-
$F_p$) during boreal winter (Fig. 4). Compared with MERRA2, the SCDM simulates well the -$F_p$ over the NH, with a
maximum and upward extension at ~60°N. The maximum over the southern hemisphere (SH) is somewhat poleward shifted.
Fig. 4b shows the seasonal evolution of –$F_p$ at 100 hPa over the mid to high latitudes of the two hemispheres. The model
captures realistically the magnitude and the seasonal cycle of –$F_p$, with a wintertime maximum in each hemisphere. However,
-$F_p$ is also somewhat underestimated, especially over the NH during summer when the wave activity flux is weak and year-
round over the SH.

Fig. 5 shows the zonal mean zonal wind for boreal winter (January-March) and austral winter (July-September). Overall, the
zonal mean zonal wind of the SCDM resembles closely the MERRA2 in terms of the positions of the subtropical and polar
night jets. However, the SCDM somewhat underestimates the strength of the polar vortex. The difference between SCDM
and MERRA2 (Fig. 5c) reveals magnitudes of 4 m s$^{-1}$ in the Arctic and 8 m s$^{-1}$ in the Antarctic. The rather barotropic
structure of the differences indicates that the negative wind biases are related to the representation of the surface drag in the
SCDM, as was suggested by WR18. There are also some negative zonal wind biases in the tropical stratosphere, perhaps
related to the inability of the SCDM to simulate the quasi-biennial oscillation (QBO).

In Fig. 6 we compare the residual mean mass streamfunction (or Brewer-Dobson circulation) between MERRA2 and the
SCDM. The stream function values of the SCDM over the NH at 100 hPa and below are generally too weak, suggesting that
the strength of the tropical upwelling from the troposphere into the stratosphere is also somewhat too small. This can be
linked to the reduced planetary wave driving (Fig. 4), which controls the strength of the residual mean circulation,
particularly in the lower stratosphere (Gerber, 2012). In the upper stratosphere, the SCDM shows a too strong and elongated
residual circulation at low latitudes and a too weak circulation at the middle latitudes over the winter hemispheres, giving the
impression of an overall too narrow Brewer-Dobson circulation. Gerber (2012) suggested that too deep Brewer-Dobson
circulation could be related to a too strong polar vortex, allowing more wave propagation into the upper stratosphere.
Although the polar vortex in our model is weaker than that in MERRA2, the EP-flux divergence in the SCDM, which
compared to MERRA2 shows more wave breaking in the subtropical upper stratosphere (not shown), is consistent with
stronger streamfunction values and thus tropical upwelling in the upper stratosphere.

### 4.2 Ozone

We now present ozone climatologies for the SCDM in terms of its spatial structure, seasonality, and interannual variability,
and connect some of the shortcomings that we find to the previously discussed biases in the dynamics. We begin with the
seasonal mean zonal mean ozone climatology (Fig. 7). The SCDM generally simulates the distribution of ozone well, with a





maximum in the tropics at ~10 hPa, and with increases from the tropics to the poles in the dynamically controlled lower
       stratosphere (below ~30 hPa). The differences (Fig. 7c) show positive biases in the middle and high latitudes during January-
       March, maximizing at ~0.9 ppmv at 30°N. This corresponds to an overestimation by 10-20% compared to MERRA2. The
       positive biases must be due to the simplified nature of the ozone scheme and internal circulation biases of the model (Fig. 6).
       We suspect that the model's circulation discrepancies are the dominant source for these biases. The ozone differences during
austral winter (July-September) resemble their northern counterparts, except for an additional negative bias over the high
       latitudes.

       The seasonal evolution of the mean total (column integrated) column ozone are shown in Fig. 8. In the Arctic, the MERRA2
       total column ozone exhibits a springtime maximum due to the enhanced meridional transport in winter (Fig. 8a). By contrast,
       Antarctic ozone undergoes a minimum in spring, resulting from anthropogenic ozone depletion. While the SCDM quite
faithfully simulates the observed evolution of the total column ozone (Fig. 8b), the ozone biases seen before (Fig. 7c) also
       imprint on the column ozone (Fig. 8c). This leads to an overestimation of ozone over most latitudes and times, except over
       Antarctica during the time of the ozone hole. Overall, our results demonstrate that the SCDM is capable of simulating a quite
       realistic global distribution and seasonal evolution of ozone.

       Strong interannual circulation variability from intermittent SSWs is an important characteristic of the northern high-latitude
stratosphere. We therefore examine next how this variability affects ozone over the two polar caps (Fig. 9). We note that, in
       contrast to the real atmosphere, the SCDM contains no interannually varying forcings (e.g., from varying sea surface
       temperatures), and that the internal dynamics of the model (e.g., SSWs) are the only source for its interannual variability.
       Considering this, we expect a reduced interannual variability in the SCDM.

       In MERRA2 over the Arctic (Fig. 9a), the variability of lower stratospheric ozone (below 10 hPa) strengthens from
November and reaches a maximum during February-March. The increased variability is associated with intermittently
       enhanced planetary wave forcing (Fig. 4b), often resulting in SSWs and associated increases in poleward ozone transports.
       The Arctic ozone variability in the SCDM (Fig. 9c) is somewhat too low during early winter, consistent with a reduced
       stratospheric wave driving during this period (Fig. 4b). But during mid-winter, the Arctic ozone variability in the SCDM
       (Fig. 9c) is somewhat too high, perhaps related to the positive ozone bias seen in the lower stratosphere. There is also a too
weak Arctic ozone variability during NH summer. Over Antarctica, the SCDM overall underestimates the ozone variability
       throughout the entire year (Fig. 9d), consistent with the negatively biased stratospheric wave driving over the SH (Fig. 4b).
       Another reason for the reduced variability over the SH is the much-simplified parameterization of heterogeneous ozone
       depletion.

       **4.3 SSW Composites**

SSWs over the NH are the most important form of stratospheric circulation variability on intraseasonal to interannual time
       scales, and the evolution of stratospheric ozone in response to SSWs is probably a key component for the interaction
       between ozone and the circulation (e.g., Butler et al., 2017; De la Cámara et al., 2018; Hocke et al., 2015; Hong and Reichler,





2021). In the following, we test the ability of the SCDM to simulate the dynamics and the transport of ozone during mid-winter (January-February) SSWs. We follow the common SSW definition by Charlton and Polvani (2007), in which the

onset of a major warming event is defined when the zonal mean zonal wind at 10 hPa and 60°N reverses from westerly to easterly. We only consider midwinter SSWs with onset dates in January or February, because SSWs during this time are strong and presumably associated with large ozone perturbations. In the 39 years of MERRA2 data (1980-2018), there occurred 13 midwinter SSWs, whereas in the 2000 years of the SCDM simulation, we find 665 midwinter SSWs. Thus, the observed and model simulated SSW frequencies are identical (0.33/year).

SSWs are usually triggered by bursts of tropospheric planetary wave activity penetrating into the stratosphere (Limpasuvan et al., 2004; Polvani and Waugh, 2004). We therefore start with examining composites of the vertical EP-flux over the life cycle of midwinter SSWs (Fig. 10a, b). MERRA2 shows that the onset of SSWs is preceded by abrupt increases in upward-propagating planetary wave activity at lead times of about 10 days; after the onset, the upward EP-fluxes are reduced for several weeks, presumably because the weakened vortex wind inhibits the upward propagation of the waves (Charney and

Drazin, 1961). The SSW composite of the SCDM captures the basic sequence of these events quite well, but the reduction of the upward EP-fluxes after the onset of SSWs is weaker than in the observations. This may be related to the non-representation of gravity waves in the SCDM, as such waves and their filtering play an important role during the recovery phase of SSWs (Limpasuvan et al., 2012).

The remaining panels of Fig. 10 are SSW composites of the zonal mean zonal wind, temperature, ozone mixing ratio, and

shortwave heating by ozone over the Arctic. The onset of SSWs is characterized by negative wind anomalies and warming temperatures over the entire stratosphere (Fig. 10c, e) (see also Limpasuvan et al., 2004). These anomalies persist particularly long in the lower stratosphere, for more than 60 days. At the same time, significant increases of ozone are observed over the Arctic polar cap due to enhanced eddy mixing and vertical transport by the residual circulation (de la Cámara et al., 2018; Hong and Reichler, 2021) (Fig. 10g), creating concurrent shortwave heating anomalies in the

stratosphere (Fig. 10i). Ozone in the chemically controller upper stratosphere is anti-correlated with temperatures (Craig and Ohring, 1958), helping to explain the positive ozone anomalies (Fig. 10g) and negative temperature anomalies (Fig. 10e) above 5 hPa.

The SCDM generally captures the observed dynamical responses to SSWs, albeit the magnitudes of simulated anomalies are sometimes weaker (Fig. 10b, d, f, h, j). For example, the temperature anomalies of the SCDM have a maximum of 10 K

during the SSW onset (Fig. 10f), while MERRA2 shows a temperature maximum of more than 14 K. The persistence and downward propagation of the zonal wind anomalies (Fig. 10d) and also the temperatures (Fig. 10f) after SSW onset also exhibit some discrepancies compared to MERRA2 (Fig. 10c, e). We believe that these discrepancies are related to the inability of the SCDM to simulate correctly the reduction in stratospheric wave driving after SSWs (Fig. 10a, b). These circulation biases impact to some extent the transport of ozone, for example in terms of more persistent ozone anomalies (Fig.

10h) and corresponding shortwave heating (Fig. 10j) in the lower stratosphere.





Overall, despite the simplicity of the SCDM, there is good agreement between the model and the reanalysis, with a reasonable simulation of the instantaneous ozone response to SSWs in both lower and upper stratosphere. When comparing the SCDM with MERRA2, one also has to take into account that the sampling uncertainty of MERRA2 (consisting of only 13 events) is large. However, we believe that the SCDM in its current form represents a useful and computationally
inexpensive tool to study the role of interactive ozone chemistry for the dynamics of the stratosphere.

## 5 Summary and Outlook

We introduce a dry dynamical core model with interactive ozone, denoted simplified chemistry-dynamical model (SCDM). With its very basic ozone-chemistry-radiation setup, the new model is located at the hierarchy between idealized GCMs and complex chemistry-climate models. SCDM is primarily targeted to investigate - in a simplified manner - the two-way
coupling between stratospheric ozone and the circulation.

SCDM is based on the dry dynamical core from GFDL, with a horizontal resolution of T42 and 40 hybrid levels. We relax temperatures using an empirically-determined seasonally varying equilibrium temperature profile to create a more realistic circulation than traditional dry dynamical cores. Ozone is transported by the model's dynamical core as a passive tracer, and a simplified linear ozone scheme introduces additional photochemical ozone tendencies based on the circulation-induced
perturbations in ozone, partial column ozone, and temperature. We employ an accurate and fast parameterization for the ozone absorption of solar radiation, which feeds back on the temperatures and the dynamics. With this setup, SCDM becomes an economical and fast tool for the study of the two-way interactions between ozone and the dynamics.

We validate the model against the MERRA2 reanalysis, a rather high benchmark given the simplicity of the SCDM. Overall, the model compares favorably against the reanalysis climatology of the stratosphere-troposphere system, both in terms of the
large-scale dynamics and the distribution of ozone. The spatial structure in the upward-propagating planetary wave activity, crucial for the residual mean circulation and the transport of ozone, is quite well simulated, but its overall magnitude is somewhat underestimated (Fig. 4). Climatological ozone is overestimated by up to 20% in the middle latitudes (Fig. 7), likely due to the biases in the residual circulation (Fig. 6) and the simplicity of the ozone scheme. Despite these biases, the seasonality (Fig. 8) and the interannual variability of ozone (Fig. 9) over both poles are well simulated.

As a proof-of-concept, we examine the dynamical variability and changes in ozone in the stratosphere and troposphere during the composite life-cycle of stratospheric sudden warmings (SSWs). The model quite faithfully simulates the well-known characteristics of SSWs, including the variations in planetary wave activity, zonal wind, temperature, and stratospheric ozone (Fig. 10). Some differences with respect to MERRA2 exist, most notably an insufficient suppression of the planetary wave activity and a too weak over-recovery of the polar vortex after the onset of SSWs. We suspect that this
problem is systemic to models with Held-Suarez forcing, related to the missing gravity wave drag and the simplified forcing of such models. Despite this, the SCDM simulates quite well all the processes relevant for the ozone-dynamics coupling in the stratosphere.

Our study contributes to an increasing diversity of idealized models, which are essential tools in the pursuit of a deeper understanding for complex atmospheric phenomena. We now envision to use SCDM for an in-depth study of the role of
interactive ozone for the variability of the coupled stratosphere-troposphere system and its associated feedbacks. A future enhanced version of the model will have an updated version of the ozone model, a parameterization for gravity waves, and an enhanced radiation scheme that also considers longwave radiation.

**Code availability**

The source code of the SCDM can be obtained at https://doi.org/10.5281/zenodo.4780888.

**Data availability**

MERRA-2 reanalysis data are available online via NASA's Goddard Earth Sciences Data and Information Services Center archive (https://gmao.gsfc.nasa.gov/reanalysis/MERRA-2/data_access/, Bosilovich et al., 2015).

**Author contribution**

HH developed the model, performed the simulations and the analysis, and wrote the manuscript. TR designed the study,
provided guidance in the interpretation of the results, and reviewed the manuscript.

**Competing interests**

The authors declare that they have no conflict of interest.

**Acknowledgements**

We thank the National Science Foundation under Grant 1446292 and the Department of Atmospheric Sciences at the
University of Utah for their support. The use of the computing infrastructure from the Center for High Performance Computing at the University of Utah is gratefully acknowledged. We also thank NASA for providing the MERRA2 reanalysis and Daniel Cariolle for providing the coefficients of their ozone scheme.

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

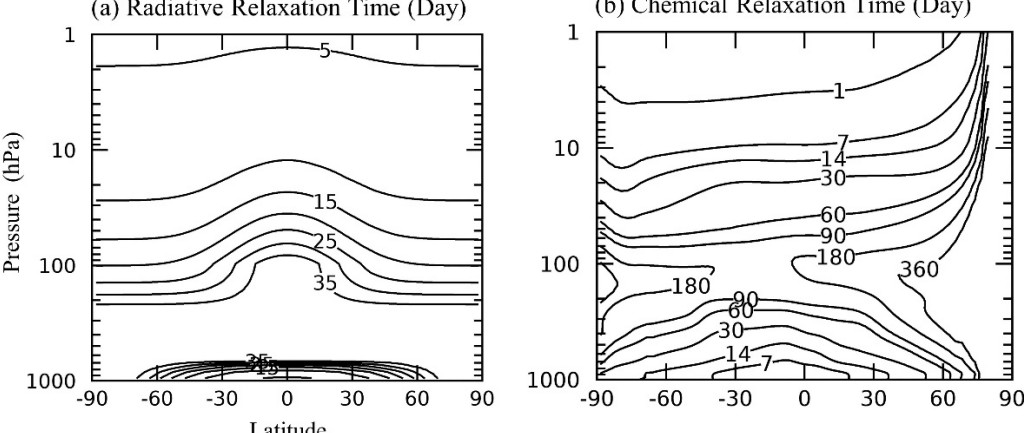

**Figure 1. Latitude-height cross-sections of relaxation times (days). (a) is the temperature (Newtonian) relaxation time $\tau$, and (b) is the chemical relaxation time for ozone ($-1/c_1$) in February.**


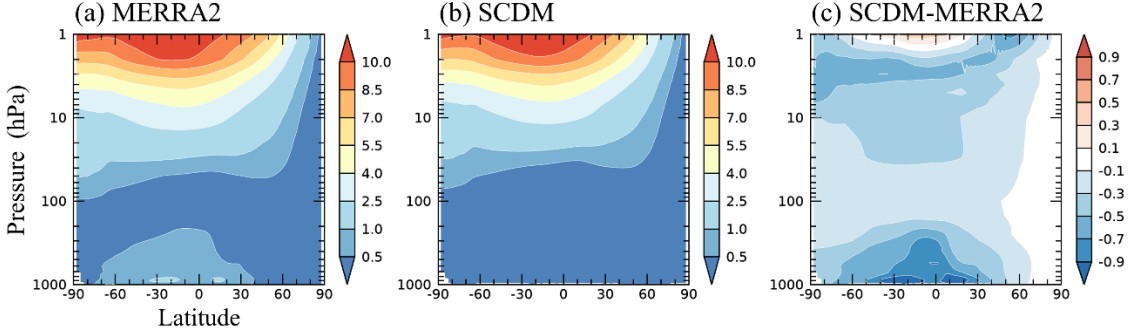

**Figure 2. January-March zonal mean shortwave heating rate (K day$^{-1}$) climatology over 1980-2018. Shown are latitude-height** 445 **cross-sections for (a) MERRA2, (b) SCDM, and (c) the difference between SCDM and MERRA2.**







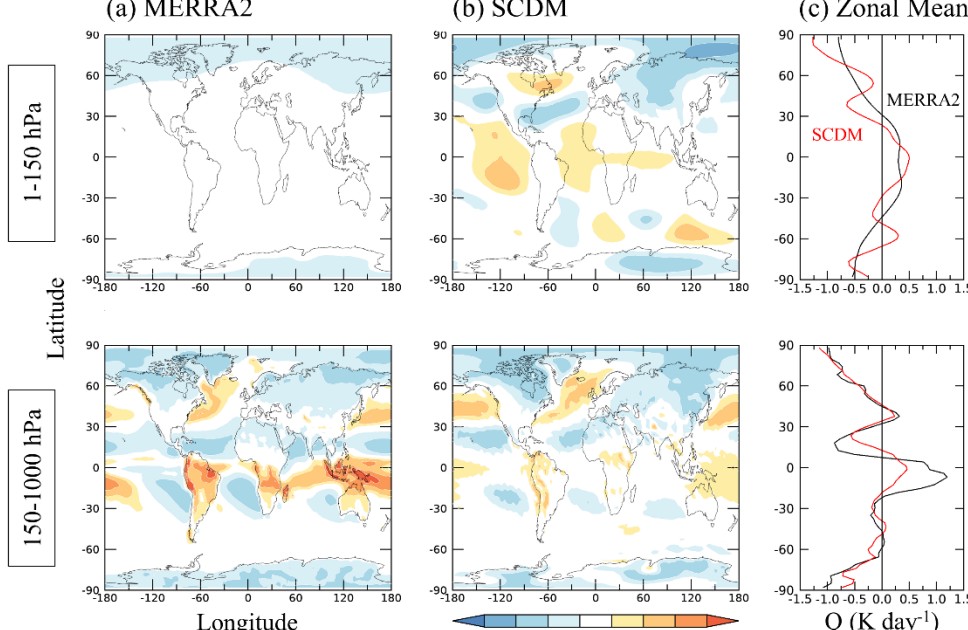

**Figure 3. January-March diabatic heating rate $Q_{clm}$ (K day$^{-1}$) for (a) MERRA2, (b) SCDM, and (c) zonal means of (a) and (b). Shown are vertical averages over (top) the stratosphere (1-150 hPa) and (bottom) the troposphere (150-1000 hPa). The MERRA2 diabatic heating data is obtained from the temperature tendency due to physics.**









(a) JFM Vertical Component of EP-Flux ($10^5$ kg m s$^{-4}$)

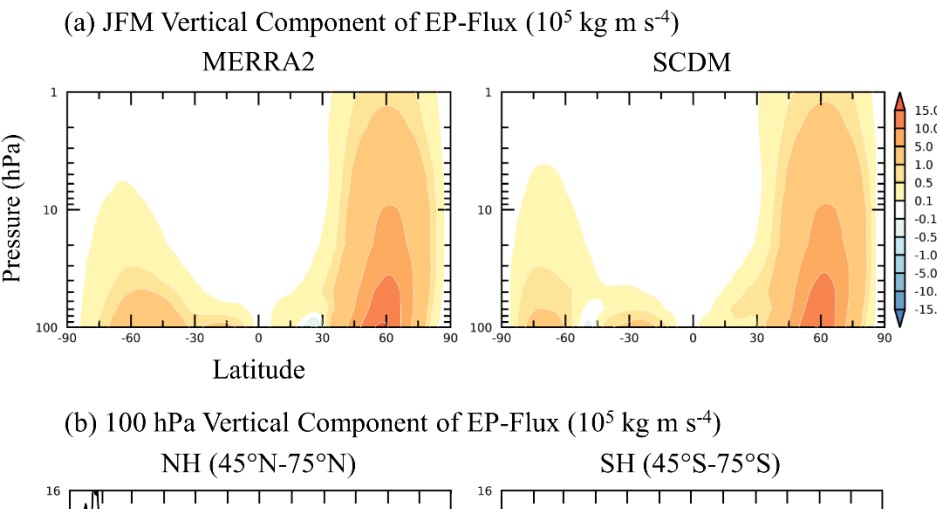

(b) 100 hPa Vertical Component of EP-Flux ($10^5$ kg m s$^{-4}$)

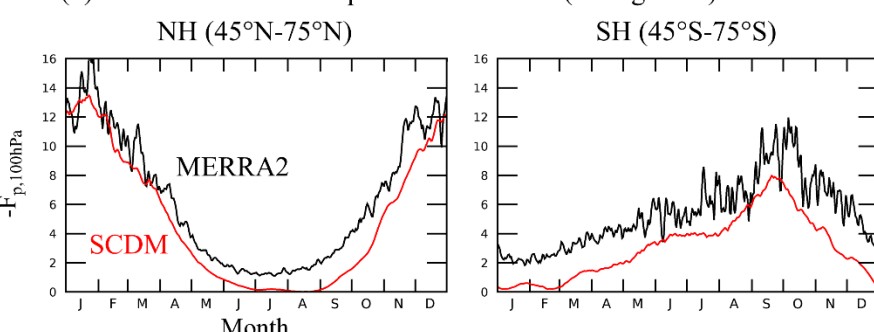

**Figure 4. Vertical EP-flux component ($10^5$ kg m s$^{-4}$). Shown are (a) latitude-height cross-sections for January-March means and (b) daily climatologies averaged over the NH (45ºN-75ºN) and the SH (45ºS-75ºS) at 100 hPa.**



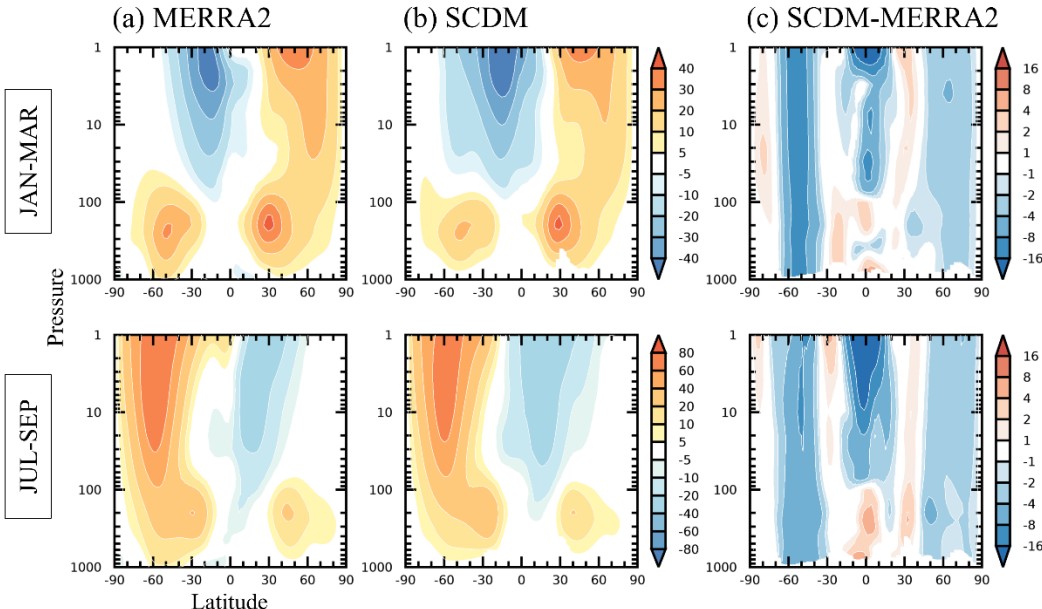

**Figure 5. Latitude-height cross-sections of zonal mean zonal wind (m s⁻¹) for (a) MERRA2, (b) SCDM, and (c) SCDM minus MERRA2. Shown are seasonal climatologies for boreal winter (January-March) and austral winter (July-September). Shading in Fig. 5c passes statistical significance at the 95% level.**

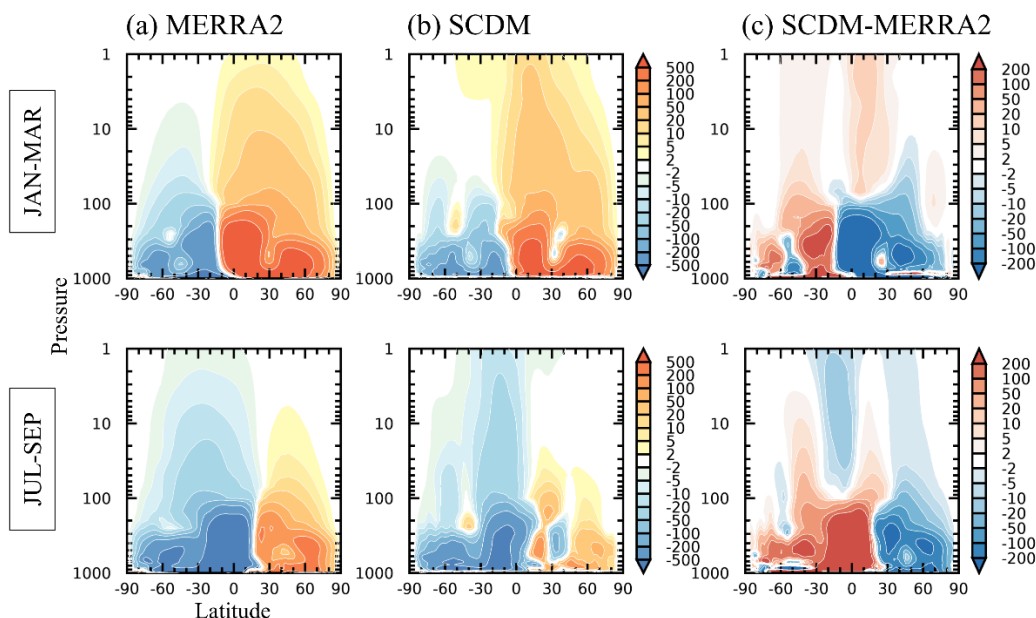

**Figure 6.** As Fig. 5, but for the residual mean mass streamfunction ($10^6$ kg s$^{-1}$).


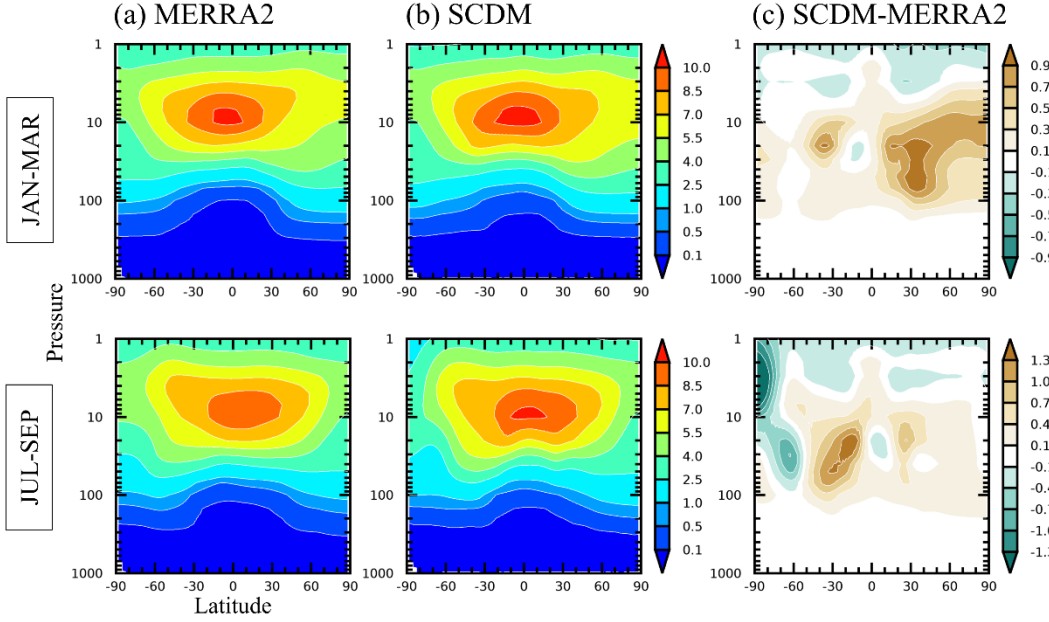

**Figure 7.** As Fig. 5, but for the zonal mean volume ozone mixing ratio (ppmv).




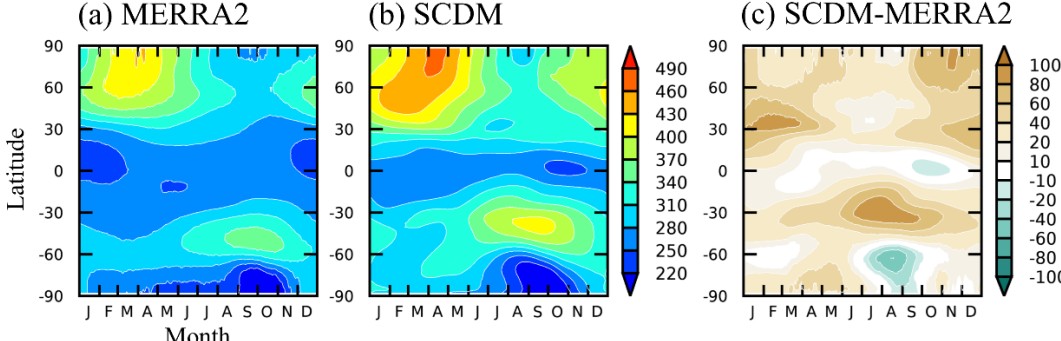

**Figure 8. Time-latitude cross-sections of daily total column ozone (DU) for (a) MERRA2, (b) SCDM, and (c) SCDM minus MERRA2. Shading in Fig. 8c passes statistical significance at the 95% level.**


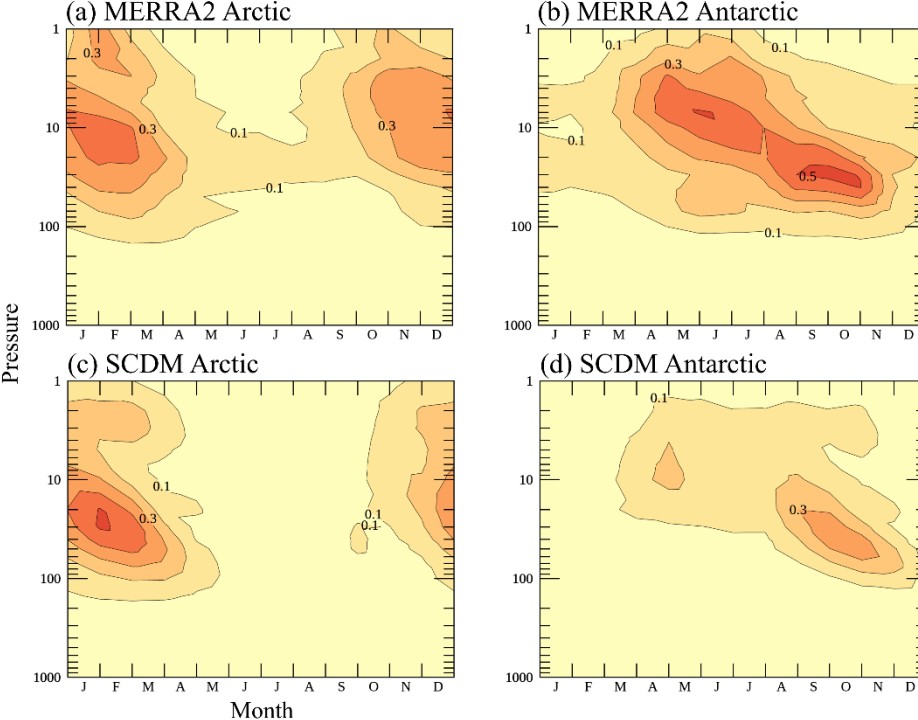

**Figure 9. Interannual ozone variability (ppmv) by month for (a-b) MERRA2 and (c-d) the SCDM. Shown are polar cap averages over (a, c) the Arctic (60°N-90°N) and (b, d) the Antarctic (60°S-90°S).**


**Figure 10. SSW composite for (a-b) the vertical EP-flux ($10^4$ kg m s$^{-4}$) and zonal mean (c-d) zonal wind $U$ (m s$^{-1}$), (e-f) temperature $T$ (K), (g-h) ozone mixing ratio $O_3$ (ppmv), and (i-j) shortwave heating by ozone $Q_{SW}$ ($10^{-3}$ K day$^{-1}$). Shown are anomalies for (a-b) 40°N-80°N, (c-d) 60°N, and (e-j) 60°N-90°N. Contours indicate statistical significance at 95% levels.**
