# Peer review of "A Simplified Chemistry-Dynamical Model"

_Geoscientific Model Development, 2021_

## Author Response (AR1)

Response to Community comment on "A Simplified Chemistry-Dynamical Model"
by Hao-Jhe Hong and Thomas Reichler, Geosci. Model Dev. Discuss.,
https://doi.org/10.5194/gmd-2021-149-CC1, 2021

*Dear Authors,*
*we have a minor remark to your citation of the "Romanowsky et al., 2019" study.*
*We are developing the fast stratospheric ozone chemistry scheme (SWIFT) used in*
*the study of Romanowsky et al., 2019.*

*In your introduction you state:*
*"Some modelling studies have already suggested a stronger stratosphere-*
*troposphere interaction when interactive ozone is introduced in their simulations*
*(Haase and Matthes, 2019; Li et al., 2016; Lin and Ming, 2021; Lin et al., 2017;*
*Romanowsky et al., 2019), but these studies were based on complex chemistry-*
*climate models that have a large computational burden and are often difficult to*
*understand."*

*We think, the SWIFT module can not be described as a "complex chemistry-*
*climate" model.*
*Its intention is to be fast and numerically efficient, therefore it is not complex and*
*has a very low computational burden.*
*SWIFT is coupled to GCMs for the same purpose as the Cariolle or Linoz schemes.*
*In the current state of SWIFT, it only treats ozone and thus provides an interactive*
*ozone layer to the GCM (e.g. replacing prescribed ozone from climatologies). So in*
*our view it does not full fill the criteria to be a full "chemistry-climate model "*
*which may include full stratospheric and trotpospheric chemistry.*

*For further information on the SWIFT model please refer to:*
*Wohltmann, I., R. Lehmann, and M. Rex. "Update of the Polar SWIFT Model for*
*Polar Stratospheric Ozone Loss (Polar SWIFT Version 2)." Geosci. Model Dev. 10,*
*no. 7 (July 13, 2017): 2671–89. https://doi.org/10.5194/gmd-10-2671-2017.*
*Kreyling, D., I. Wohltmann, R. Lehmann, and M. Rex. "The Extrapolar SWIFT*
*Model (Version 1.0): Fast Stratospheric Ozone Chemistry for Global Climate*
*Models." Geosci. Model Dev. 11, no. 2 (March 1, 2018): 753–69.*
*https://doi.org/10.5194/gmd-11-753-2018.*

*Don't hesitate to write us if you have more questions (daniel.kreyling@awi.de).*

***Kind Regards, Daniel Kreyling***

Dear Daniel,

Thank you for your comment. We apologize for characterizing your model "SWIFT" as a complex chemistry model. In the revised version of our manuscript, we will correct our mistake.

With kind regards,
Hao-Jhe

We revised our text in the manuscript at line 35 and line 42:

L35: *"Ozone has long been realized as an important atmospheric trace gas that influences the trend of the Southern Annular Mode (SAM) and the strength of the Antarctic polar vortex through its radiative heating (Gillet and Thompson, 2003; **Lin et al., 2017**; Randel and Wu, 1999; Seviour et al., 2017; Son et al., 2010; Thompson and Solomon, 2002)."*

L42: *"Some modelling studies have already suggested a stronger stratosphere-troposphere interaction when interactive ozone is introduced in their simulations (Haase and Matthes, 2019; Li et al., 2016; Lin and Ming, 2021; Romanowsky et al., 2019), but these studies were based on full climate models coupled to chemistry modules of different complexities. Most of these models have a large computational burden and are often difficult to understand."*

Response to Chief editor comment on "A Simplified Chemistry-Dynamical Model" by Hao-Jhe Hong and Thomas Reichler, Geosci. Model Dev. Discuss., https://doi.org/10.5194/gmd-2021-149-CEC1, 2021

**Dear authors,**
**in my role as Executive editor of GMD, I would like to bring to your attention our Editorial version 1.2: https://www.geosci-model-dev.net/12/2215/2019/**

**This highlights some requirements of papers published in GMD, which is also available on the GMD website in the 'Manuscript Types' section: http://www.geoscientific-model-development.net/submission/manuscript_types.html**

**In particular, please note that for your paper, the following requirement has not been met in the Discussions paper:**
- **"The main paper must give the model name and version number (or other unique identifier) in the title."**

**Please add a version number for SCDM in the title upon your revised submission to GMD.**

**Yours,**
**Astrid Kerkweg**

Dear Astrid,

Thank you for pointing this out to us. In the revised version of the paper, we will use the following title: "A Simplified Chemistry-Dynamical Model (SCDM V1.0)".

Best,
Hao-Jhe

In the revised manuscript, we modified the title as *"A Simplified Chemistry-Dynamical Model (SCDM V1.0)"*.

Response to Referee #1 comment on "A Simplified Chemistry-Dynamical Model" by Hao-Jhe Hong and Thomas Reichler, Geosci. Model Dev. Discuss., https://doi.org/10.5194/gmd-2021-149-RC1, 2021

*Overall, I find this paper well-written and clear and find that SCDM adds a unique element to the dry dynamical modelling hierarchy, allowing for examination of the feedbacks between interactive stratospheric ozone and circulation.*

*My main comment is that given that the original Teq includes the climatological effects of ozone, it would have been nice to see how this version of the model differs. For example, it would have been interesting to compare SCDM with just climatological ozone versus SCDM with interactive ozone and compare the SSW diagnostics. This would have provided some proof-of-concept that interactive ozone can cause differences in SSW evolution, for example.*

Yes, we agree, the main purpose of the SCDM is to investigate the influence of interactive ozone on the circulation. And indeed, related work is already underway, but we very much prefer to publish the outcomes in a separate paper. To give the reviewer some idea of the impacts of interactive ozone, we show in Fig. R1 the differences in the interannual variability of zonal-mean zonal wind and temperature from two simulations. The first simulation is SCDM with interactive ozone, as described in the present manuscript, and the second simulation (PrO3) is from the SCDM but prescribing the climatological ozone from the first simulation. Fig. R1 shows that interactive ozone leads to significant increases in variability, mainly in April and May. The timing is to be expected since for ozone to have temperature effect, sun light is required.

Next, to understand whether interactive ozone influences the evolution of SSWs, we compare in Fig. R2 temperature composites of February SSWs. The difference between SCDM and PrO3 (panel c) suggests a more persistent temperature anomaly in the lower stratosphere when ozone is interactive. As said above, to achieve a more in-depth understanding for the role of interactive ozone, we already have and will perform further analysis, and we intend to publish the results in a separate paper. We now mention this in the conclusion of the manuscript at line 309:

L309: *"In upcoming work, we will use SCDM for an in-depth study of the role of interactive ozone for the variability of the coupled stratosphere-troposphere system and its associated feedbacks."*

[Figure]

**Figure. R1.** Time-height cross-sections of the interannual variability for (a) zonal-mean zonal wind at 60ºN and (b) zonal-mean temperature averaged over 60ºN-90ºN. Shown are differences between the SCDM run (interactive ozone) and the PrO3 run (prescribed three-dimensional ozone climatology from the SCDM run). Contours represent statistical significance of the difference at the 95% level using an F-test.

[Figure]

**Figure. R2.** February SSW composite for polar cap averaged (60ºN-90ºN) temperatures. Shown are results for (a) SCDM (interactive ozone), (b) PrO3 (prescribed ozone), and (c) SCDM-PrO3. Contours in (a) and (b) represent statistical significance of the anomaly at the 95% level using a two-tail Student t-test and in (c) indicate that the differences between SCDM and PrO3 are significant at the 95% level according to an F-test.

*Minor Comments:*

*1. Line 32: I think several other papers using the dry dynamical core have employed realistic topography (e.g. Wu and Smith, 2016)*

We added this reference at line 32:

L32: *"...or actual topography (Wu and Reichler, 2018; Wu and Smith, 2016)..."*

**2. Figure 3b: Is it possible that the diabatic heating differences between MERRA2 and SCDM in the tropics are related to an unrepresented QBO? This was noted in the text regarding the tropical zonal wind differences.**

The stratospheric diabatic heating terms shown in Fig. 3 (top) represent pressure-weighted averages from 1-150 hPa, and therefore include some tropospheric contributions in the tropics. To avoid this, we now modify our analysis and only integrate from 1-70 hPa, which is everywhere in the stratosphere. The new result (Fig. R3), which is also included in the new manuscript (Fig. 3, top), suggests that SCDM simulates reasonably well the diabatic heating in the stratosphere. Some differences exist at low latitudes, which we believe are related to errors in tropical ascent and the correction of the resulting adiabatic heating and temperature errors by the iterative procedure. Major differences, however, remain in the tropical troposphere (Fig. 3, bottom), which we believe are primarily related to a too weak Hadley circulation in the SCDM and missing latent heating from convective activity in the inner tropics (Fig. 6c).

We do not believe that the differences between SCDM and MERRA2 are related to the missing QBO in the model, because the oscillatory nature of the QBO would lead to zero anomalies in the long-term mean. Also, the QBO circulation does not extend below 100 hPa and is thus unlikely to influence the tropospheric (150-1000 hPa) diabatic heating term. In the revised manuscript, we replace the top of Fig. 3 with Fig. R3 and modify the corresponding text at line 169:

L169: *"The diabatic heating in the stratosphere from SCDM also agrees well with MERRA2 (Fig. 3, top). The major discrepancies occur in low latitudes, which we believe are related to errors in tropical ascent and the correction of the resulting adiabatic heating and temperature errors by the iterative procedure. In the zonal mean, MERRA2 and the model are in rather good agreement."*

[Figure]

**Figure. R3.** January-March diabatic heating rate $Q_{clm}$ (K day-1) for (a) MERRA2, (b) SCDM, and (c) zonal means of (a) and (b). Shown are vertical averages over the stratosphere (1-70 hPa). The MERRA2 diabatic heating data are estimated from the temperature tendencies due to physics.

*3. Figures 9 and 10: The authors suggest that the lack of gravity wave representation may play a role in the differences in SSWs, but these figures may also suggest that the relaxation times in SCDM are not tuned quite right. Do you think that the Newtonian and/or chemical relaxation times need to be adjusted for this configuration of the model? Did the authors test retuning the relaxation times?*

The reviewer makes a very good point here, but we wanted to be consistent with previous work (Jucker et al. 2014; WR18) and keep the dynamical relaxation time at its original values. Also, when considering the SSW evolution in temperature (Fig. 10f), the time scale of SCDM in the lower stratosphere is fairly close to that of MERRA2 (Fig. 10e). We agree that the SCDM time scale looks much longer in the zonal wind composite (Fig. 10d), especially in the upper stratosphere. We note that most idealized models have difficulty simulating the typical "over-recovery" of the polar vortex after onset, since gravity wave filtering plays a crucial role in modulating the winds in the stratosphere. The missing over-recovery, then, has major implications for all fields, in particular in the middle and upper stratosphere, and gives the appearance of an overly persistent model. In the manuscript, we now write at line 310:

L310: "*Possible future model enhancements will include an updated version of the ozone parameterization, a parameterization for gravity waves, and an enhanced radiation scheme that also considers longwave radiation. We will also consider retuning the Newtonian relaxation time scale to bring the model in even better agreement with the observations.*"

Response to Referee #2 comment on "A Simplified Chemistry-Dynamical Model" by Hao-Jhe Hong and Thomas Reichler, Geosci. Model Dev. Discuss., https://doi.org/10.5194/gmd-2021-149-RC2, 2021

*This paper describes a simplified chemistry-dynamical model (SCDM), which consists of a dry dynamical core, a simple linear ozone scheme, and an ozone shortwave parameterization. In the SCDM, stratospheric chemistry couples with dynamics through ozone shortwave absorption. Ozone concentrations are determined by transport and photochemical tendencies calculated from the linear ozone scheme. Changes in ozone affect temperature and dynamics through shortwave absorption. SCDM climatology is validated against MERRA2 in terms of ozone, shortwave radiation, and dynamical fields. In addition, ozone and dynamical variability associated with the Arctic stratospheric sudden warming (SSW) events in SCDM are compared to that in MERRA2. Overall, the SCDM simulates reasonably well the climatology and variability of stratospheric ozone and dynamics.*

*Interactive stratospheric ozone chemistry has been shown to play a key role in stratosphere-troposphere coupling. Given its simplicity and computational efficiency in relative to the comprehensive chemistry-climate models, the SCDM provides a potentially useful tool to understand the stratospheric chemical-dynamical coupling. The paper is well written. The description of the model is clear. I recommend publication after my comments are addressed.*

*Comments:*

*The SCDM simulates well the Arctic SSWs. Does it also capture the Antarctic temperature and wind variability? I suggest adding a figure like Figure 9 for wind and temperature.*

We now add to the paper an analysis for temperature and wind variability over the Arctic and Antarctic. The new Fig. 9 (or Fig. R4 below) reveals that the model also captures reasonably well the circulation and ozone variability over the Antarctic. In the revised manuscript we add a brief discussion of the new figure. Please see line 225 and 229 in the revised manuscript, where we write:

L225: *"...We therefore examine next how this variability affects the circulation and ozone over the two polar caps (Fig. 9)."*

L229: *"In MERRA2 over the Arctic (Fig. 9a, e, i), the variability of lower stratospheric zonal wind, temperature, and ozone (below 10 hPa) strengthens from November and reaches a maximum during February-March. The increased variability is associated with intermittently enhanced planetary wave forcing (Fig. 4b), often resulting in SSWs and associated increases in poleward ozone transports. The Arctic temperature and ozone variability in the SCDM (Fig. 9f, j) is somewhat too low during early winter, consistent with a reduced stratospheric wave driving during this period (Fig. 4b). But during mid-winter, the Arctic ozone variability in the SCDM (Fig. 9j) is somewhat too high, perhaps related to the positive ozone bias seen in the lower stratosphere. There is also a too weak Arctic ozone variability during NH summer. Over Antarctica, the SCDM overall somewhat underestimates the temperature and ozone variability throughout the entire year (Fig. 9h, l), consistent with the negatively biased stratospheric wave driving over the SH (Fig. 4b). Another reason for the reduced variability over the SH is the much-simplified parameterization of heterogeneous ozone depletion."*

[Figure]

**Figure. R4.** Interannual variability of zonal-mean (a-d) zonal wind, (e-h) temperature, and (i-l) ozone. The interannual variability is estimated by standard deviation of $T$ and $O_3$ over the Arctic (60ºN-90ºN) and the Antarctic (60ºS-90ºS) and by standard deviation of $U$ at 60ºN and 60ºS. Results are shown for (a, c, e, g, i, k) MERRA2 and (b, d, f, h, j, l) SCDM.

*Following up the above question, does interactive ozone improve the simulation of stratospheric dynamical variability in SCDM? In order to answer this question, an additional simulation without interactive ozone is required. If such simulation is available, I suggest adding a section to discuss this topic.*

The related work for addressing the impact of interactive ozone is underway, but we very much prefer to publish the new outcomes in a more comprehensive separate paper. To give the reviewer some idea of the impacts of interactive ozone, we show in Fig. R1 the differences in the interannual variability of zonal-mean zonal wind and temperature from the two simulations. The first simulation is SCDM with interactive ozone, as described in the present manuscript, and the second simulation (PrO3) is from the SCDM but prescribing the climatological ozone from the first simulation. Fig. R1 shows that interactive ozone leads to significant increases in variability, mainly in April and May. The timing is to be expected since for ozone to have temperature effect, sun light is required.

Next, to understand whether interactive ozone influences the evolution of SSWs, we compare in Fig. R2 temperature composites of February SSWs. The difference between SCDM and PrO3 (panel c) suggests a more persistent temperature anomaly in the lower stratosphere when ozone is interactive. As said above, to achieve a more in-depth understanding for the role of interactive ozone, we already have and will perform further analysis, and we intend to publish the results in a separate paper. We now mention this in the conclusion of our manuscript at line 309, where we write:

L309: *"In upcoming work, we will use SCDM for an in-depth study of the role of interactive ozone for the variability of the coupled stratosphere-troposphere system and its associated feedbacks."*

[Figure]

**Figure. R1.** Time-height cross-sections of the interannual variability for (a) zonal-mean zonal wind at 60ºN and (b) zonal-mean temperature averaged over 60ºN-90ºN. Shown are differences between the

SCDM run (interactive ozone) and the PrO3 run (prescribed three-dimensional ozone climatology from the SCDM run). Contours represent statistical significance of the difference at the 95% level using an F-test.

[Figure]

**Figure. R2.** February SSW composite for polar cap averaged (60ºN-90ºN) temperatures. Shown are results for (a) SCDM (interactive ozone), (b) PrO3 (prescribed ozone), and (c) SCDM-PrO3. Contours in (a) and (b) represent statistical significance of the anomaly at the 95% level using a two-tail Student t-test and in (c) indicate that the differences between SCDM and PrO3 are significant at the 95% level according to an F-test.

***Line 310. I wonder what specific topics the authors plan to investigate (or have already investigated). And how do they plan to use the SCDM model to get a better understanding of these topics?***

The main purpose of the SCDM is to study the role of interactive ozone for the variability of the coupled stratosphere-troposphere system and its associated feedbacks, especially for SSWs. As mentioned above, we address this by comparing two simulations using SCDM. The first simulation uses interactive ozone, and the second simulation uses a prescribed ozone climatology from the first simulation. We now mention this at line 310, where we write:

[revised manuscript text omitted]